# Influence of the 135 bp Intron on Stilbene Synthase *VaSTS11* Transgene Expression in Cell Cultures of Grapevine and Different Plant Generations of *Arabidopsis thaliana*

Konstantin V. Kiselev *, Zlata V. Ogneva, Olga A. Aleynova ⓘ, Andrey R. Suprun ⓘ, Alexey A. Ananev ⓘ, Nikolay N. Nityagovsky and Alexandra S. Dubrovina ⓘ

Laboratory of Biotechnology, Federal Scientific Center of the East Asia Terrestrial Biodiversity, Far Eastern Branch of the Russian Academy of Sciences, 690022 Vladivostok, Russia
* Correspondence: kiselev@biosoil.ru; Tel.: +8-423-2310410; Fax: +8-4232-310193

**Abstract:** Modern plant biotechnology often faces the problem of obtaining a stable and powerful vector for gene overexpression. It is known that introns carry different regulatory elements whose effects on transgene expression have been poorly studied. To study the effect of an intron on transgene expression, the stilbene synthase 11 (*VaSTS11*) gene of grapevine *Vitis amurensis* Rupr. was selected and overexpressed in grapevine callus cell cultures and several plant generations of *Arabidopsis thaliana* as two forms, intronless *VaSTS11c* and intron-containing *VaSTS11d*. The *STS* genes play an important role in the biosynthesis of stilbenes, valuable plant secondary metabolites. *VaSTS11d* contained two exons and one intron, while *VaSTS11c* contained only two exons, which corresponded to the mature transcript. It has been shown that the intron-containing *VaSTS11d* was better expressed in several generations of transgenic *A. thaliana* than *VaSTS11c* and also exhibited a lower level of cytosine methylation. As a result, the content of stilbenes in the *VaSTS11d*-transgenic plants was much higher than in the *VaSTS11c*-transgenic plants. Similarly, the best efficiency in increasing the content of stilbenes was also observed in grapevine cell cultures overexpressing the intron-containing *VaSTS11d* transcript. Thus, the results indicate that an intron sequence with regulatory elements can have a strong positive effect on both transgene expression level and its biological functions in plants and plant cell cultures.

**Keywords:** piceid; plant cell cultures; resveratrol; transcription factors; transgenic cells; vitis

## 1. Introduction

Modern plant biotechnology and genetic engineering often face the problem of obtaining a stable and powerful expression vector [1]. Plant organisms perceive the introduced transgene as foreign DNA and, with the help of epigenetic mechanisms, reduce the level of transgene expression [2]. Thus, an important challenge in the field of biotechnology is the vector design for the stable and strong expression of plant transgenes in crop plant tissue improvements and ornamental plant modifications.

Commonly, plant transgenes contain no introns, a feature shared with transposons, which are also prime targets for gene silencing [2]. Given that introns are very common in plant genes but are often lacking in introduced transgenes and plant transposons, it has been hypothesized that introns may contribute to gene silencing suppression [3]. It has been found that small RNA libraries of *Arabidopsis thaliana* were strongly enriched for exon sequences derived from intronless genes [3]. According to the study, compared to intronless transcripts, spliced transcripts possess a reduced ability to act as a substrate for an RNA-dependent RNA polymerase (RDR) and are less effective substrates for gene silencing. Therefore, it is possible that a mechanism for the intron-mediated suppression of gene silencing exists in plants and other organisms [3].

According to recent findings, aberrant RNAs derived from intronless transgenes and endogenes are eliminated via the RNA-directed RNA polymerase 6 (RDR6)/DICER-LIKE 4 (DCLs) pathway [4]. However, it has been assumed that aberrant RNAs derived from intron-containing transgene and endogene sequences are preferentially targeted to exonucleolytic RNA decay pathways [3–5].

While gene intronic regions do not encode proteins, they may encode for non-protein-coding RNA (ncRNAs), which perform gene regulation functions [6]. Using nematode *Caenorhabditis elegans*, it has been shown that such intron-derived ncRNAs may play an important role in the regulation of gene expression [6]. It has been shown that the products of the *lin-4* gene of *C. elegans* are small ncRNAs derived from introns. They regulate the expression of the *lin-14* gene via an antisense RNA-RNA interaction, which participates in the regulation of the postembryonic development of these animals [6]. Moreover, introns may contain regulatory elements acting as binding sites for transcription factors that can regulate transgene expression [7].

In this study, using several generations of Arabidopsis and grapevine callus cell cultures, we analyzed the expression of the grapevine stilbene synthase *VaSTS11* gene, as a representative of the multigenic *STS* family with the shortest intron in *Vitis amurensis* Rupr. Stilbenes are a group of plant phenolic compounds with antimicrobial activities and a wide range of health-beneficial effects [8–11]. Stilbenes are synthesized via the phenylpropanoid pathway by a broad range of unrelated plant families [11–13]. Stilbene synthase (STS; EC 2.3.1.95) is known as a key enzyme in stilbene biosynthesis, catalyzing the formation of simple monomeric stilbenes (e.g., resveratrol) from coenzyme A-esters of cinnamic acid derivatives and three malonyl-CoA units in a single reaction [14]. Then, t-resveratrol and other monomeric stilbenes may be metabolized to form other stilbenes, such as pterostilbene via the methylation of resveratrol by resveratrol O-methyltransferase or Romt [15]; piceid via resveratrol glycosylation by glucosyltransferases [16]; or viniferins via oxidation by polyphenol oxidase or PPO [17].

In this study, we analyzed the difference in the expression and cytosine methylation of the intronless transgene *VaSTS11c* and the intron-containing transgene *VaSTS11d* that was transgenic homozygous for the transgene *A. thaliana* in the fourth ($T_4$) and seventh ($T_7$) plant generations. The level of expression of the two *VaSTS11* transgene forms and the stilbene production level were also analyzed in callus cell cultures of *V. amurensis*.

## 2. Results and Discussion

### 2.1. VaSTS11 Transgene Expression in the VaSTS11-Transgenic A. thaliana

For quantitative RT-PCR (RT-qPCR), the same primers were used for the amplification of the *VaSTS11d* and *VaSTS11c* transcripts based on their identical exon sequences. Therefore, we can compare the expression levels of *VaSTS11d* and *VaSTS11c* in different plant generations of the transgenic Arabidopsis using the two independent plant lines for *VaSTS11d* and *VaSTS11c* (Figure 1) obtained by the floral dip method [18]. The highest transgene expression level was detected for the intron-containing *VaSTS11d* transgene in the $T_4$ plants of both ST11d-1 and ST11d-2 plant lines (Figure 1). The intron-containing *VaSTS11d* expression was considerably higher than the expression of the intronless *VaSTS11c* in the $T_4$ generation in both transgenic plant lines. In the $T_7$ generation, the *VaSTS11d* expression level in the ST11d1 and ST11d-2 plant lines considerably decreased (by 3.6–4.1-fold) compared to $T_4$ and reached the values of the intronless *VaSTS11c* in one of the transgenic lines (Figure 1). A decrease in the transgene expression level was also detected for *VaSTS11c* in the $T_7$ *VaSTS11c*-transgenic plant generation as compared to the $T_4$ *VaSTS11c*-transgenic plants. The *VaSTS11c* expression level decreased by 1.2–2.8 times, so the *VaSTS11c* mRNA level in the $T_7$ plants did not significantly differ from the unspecific amplification in WT and KA0 control plants (Figure 1).

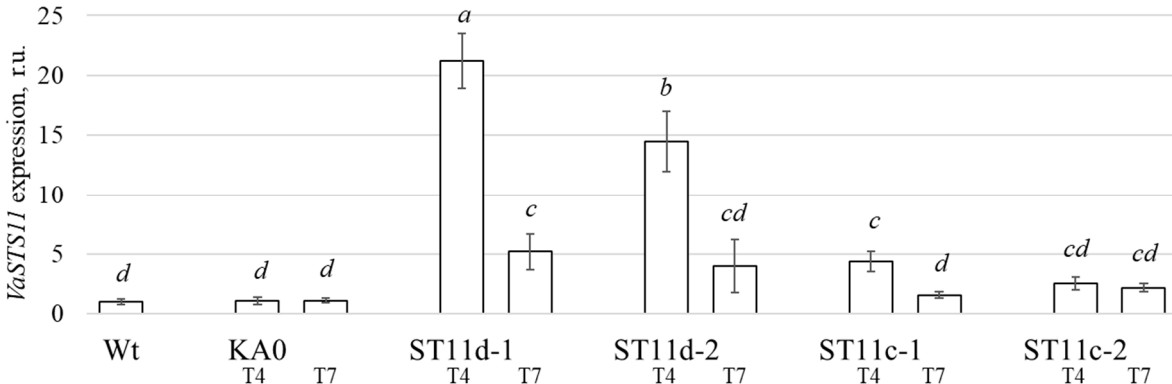

**Figure 1.** Quantification of the *VaSTS11d* and *VaSTS11c* transgene expression levels in transgenic *Arabidopsis thaliana* of $T_4$ and $T_7$ generation performed by quantitative RT-PCR. Wt—wild-type *A. thaliana* plants; KA0—the control KA0 *A. thaliana* plants transformed with the vector harboring only the *npt*II selective marker; ST11d-1, ST11d-2—two transgenic lines of *A. thaliana* plants transformed with the intron-containing *VaSTS11d*; ST11c-1, 2—*A. thaliana* plants transformed with the intronless *VaSTS11c*; r.u.—related units; $T_4$ and $T_7$—*VaSTS11d*- and *VaSTS11c*-transgenic *A. thaliana* in the fourth ($T_4$) and seventh ($T_7$) generations. The data are presented as mean $\pm$ SE (two independent experiments with eight technical replicates). Means followed by the same letter were not different using Student's *t* test, where $p < 0.05$ was considered to be statistically significant.

*2.2. Stilbene Content in the VaSTS11-Transgenic A. thaliana*

Then, the stilbene levels in the obtained *VaSTS11d*- and *VaSTS11c*-transgenic plant lines were analyzed using HPLC. Recent investigations show that *trans*-resveratrol and *trans*-piceid were the prevalent stilbenes in transgenic *A. thaliana* overexpressing different *STS* genes [19]. However, in the *VaSTS11*-transgenic plant lines (ST11d-1, ST11d-2, ST11c-1, and ST11c-2), we consistently detected only *trans*-piceid (Figure 2a) and did not detect peaks of other stilbenes. The highest content of *trans*-piceid (17.9 µg/g of dry weight, Figure 2b) was detected in the $T_4$ generation of the intron-containing ST11d-1 line previously demonstrating the highest expression of the *VaSTS11* transgene (Figure 1). In the $T_7$ generation of the ST11d-1 line, the content of *trans*-piceid was considerably lower than its content in the $T_4$ generation—by 1.4 times, reaching 12.4 µg/g of dry weight (Figure 2b). For the ST11d-2 line, we also observed a decrease in stilbene content in the $T_7$ generation in comparison to $T_4$ plants, but the difference was not statistically considerable. The lowest content of stilbenes was detected in the intronless ST11c-2 line in the $T_7$ generation, only 0.3 µg/g of dry weight (Figure 2b). In all *VaSTS11*-transgenic lines, the total stilbene content positively correlated (r = 0.95) with the level of *VaSTS11* transgene expression and significantly decreased in the $T_7$ generation compared with that in the $T_4$ generation.

There were two studies where the sorghum *SbSTS1* gene was overexpressed in Arabidopsis tt4 mutants, which could not make flavonoids, resulting in the content of stilbenes reaching 600 µg/g FW (*cis*-piceid, resveratrol diglucoside, and *t*- and *cis*-resveratrol acetyl-hexosides) in Arabidopsis leaves [20,21]. This stilbene content is one of the highest levels in transgenic Arabidopsis plants. It is possible that this level may have been reached due to an excess in stilbene precursors in these plants, since the tt4 mutant line could not produce flavonoids that require the same precursors as stilbenoid compounds. The biosynthesis pathways of the flavonoids and stilbenes use the same precursors [22,23].

In other papers, transformations of wild-type *A. thaliana* plants with the grapevine *VaSTS1* and *VaSTS7* genes led to stilbene production in *A. thaliana* with stilbene content reaching 22.7 µg/g FW for *VaSTS1* and 0.1 µg/g FW for *VaSTS7* [24]. Thus, stilbene levels in the *VaSTS11*-transgenic *A. thaliana* were closer to the average stilbene levels in *STS*-transgenic *A. thaliana*.

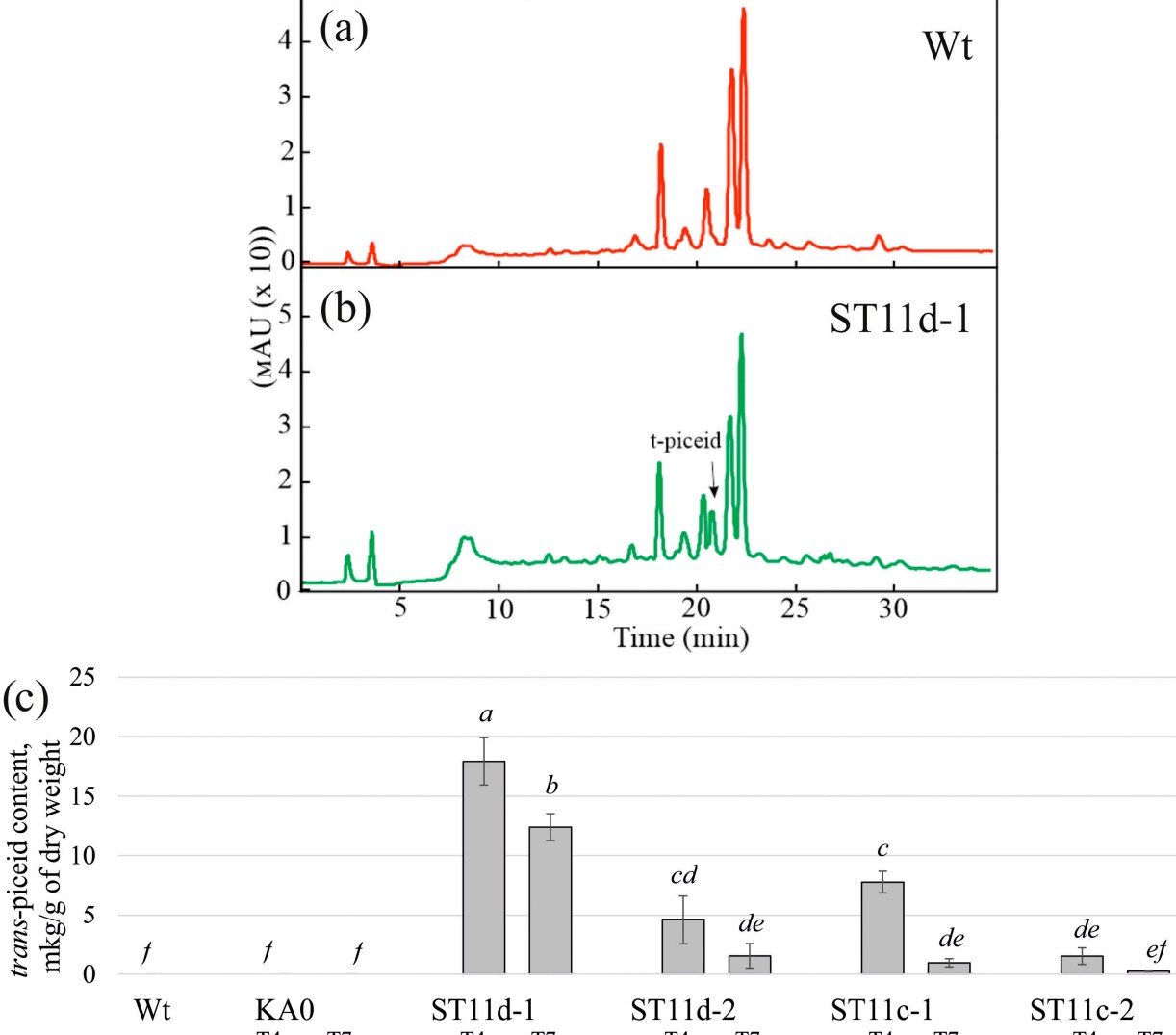

**Figure 2.** HPLC detection (**a**,**b**) and quantification (**c**) of the *trans*-piceid content in the *VaSTS11d* and *VaSTS11c* transgenic *Arabidopsis thaliana*. Wt—wild-type *A. thaliana* plants; KA0—the control KA0 *A. thaliana* plants transformed with the vector harboring only the *npt*II selective marker; ST11d-1, ST11d-2—two *A. thaliana* plant lines transformed with the *VaSTS11d* gene; ST11c-1, ST11c-2—two *A. thaliana* plant lines transformed with the *VaSTS11c* gene; $T_4$ and $T_7$—*VaSTS11d* and *VaSTS11c* expression in transgenic *A. thaliana* of the $T_4$ and $T_7$ generations. The data are presented as mean ± SE (two independent experiments with eight technical replicates). Means followed by the same letter were not different using Student's *t* test, where $p < 0.05$ was considered to be statistically significant.

### 2.3. VaSTS11 Cytosine Methylation in Transgenic A. thaliana

Using bisulfite sequencing, we analyzed the cytosine DNA methylation level of the 394 bp fragment of the 3′ end of the STS coding region in the *VaSTS11d* and *VaSTS11c* transgenes. The total level of cytosine DNA methylation in the *VaSTS11c* transgene of the $T_4$ plant generation was 1.2–1.4 times higher than the total level of cytosine *VaSTS11d* methylation in the $T_4$ generation (Figure 3). In all transgenic lines, the transgene cytosine methylation increased in plants of the $T_7$ generation in comparison with the $T_4$ plants. Moreover, it reached very high values—up to 91.1–94.2% in the intronless ST11c-1 and ST11c-2 lines (Figure 3). The data indicate that the *VaSTS11c* transgene in these plants in the $T_4$ and $T_7$ generations is high and can be indicated as hypermethylated [25].

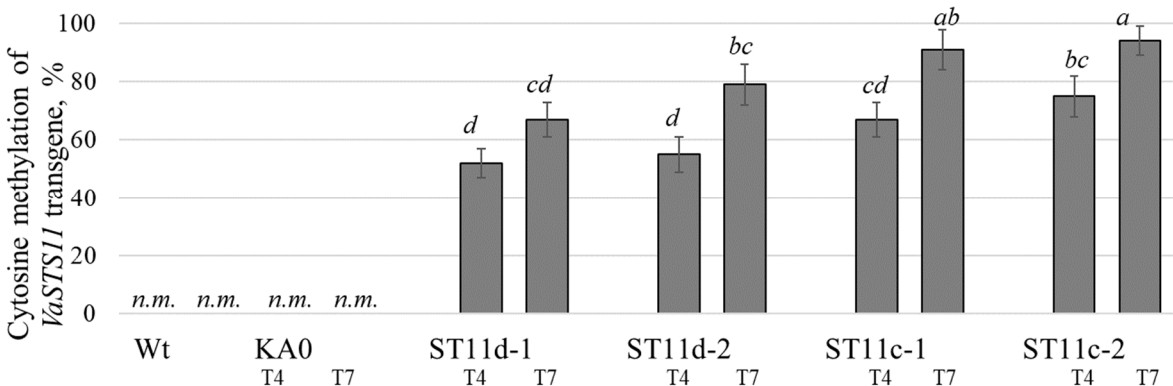

**Figure 3.** Cytosine methylation level (%) within the selected part of the protein coding regions of the *VaSTS11d* and *VaSTS11c* genes. Wt—wild-type *A. thaliana* plants; KA0—the control KA0 *A. thaliana* plants transformed with the vector harboring only the *npt*II selective marker; ST11d-1, ST11d-2—*A. thaliana* plants transformed with the *VaSTS11d* gene; ST11c-1, ST11c-2—*A. thaliana* plants transformed with the *VaSTS11c* gene; $T_4$ and $T_7$—*VaSTS11d* and *VaSTS11c* expression in transgenic *A. thaliana* of the $T_4$ and $T_7$ generations; n.m.—not measured. The data are presented as mean $\pm$ SE (two independent experiments with eight technical replicates). Means followed by the same letter were not different using Student's *t* test, where $p < 0.05$ was considered to be statistically significant.

*2.4. Search for Consensus Patterns of Plant Regulatory Sequences in the VaSTS11 Intron Using NSITE-PL*

Using the NSITE-PL plant regulatory element recognition database [7], we identified four regulatory elements in the region of 135 nucleotides of the *VaSTS11* intron: Gap box 2, EIN3 BS2, E-box, and ANAC089 BS 2 (Figures 4 and S1, Table 1).

**Table 1.** Information on the regulatory element (RE) found in the singular 135 bp *VaSTS11* intron sequences in the NSITE-PL database.

| RE Name | RE Registration Number in the NSITE-PL Database | Gene | RE Binding Factors |
|---|---|---|---|
| Gap box 2 | 1027, RSP01020 | GapB | GAPF |
| EIN3 BS2 | 2013, RSP01979 | EBF2 | EIN3 |
| E-box | 2473, RSP02439 | FT | CIB1; CIB2, CIB3, CIB4, CIB5 |
| E-box | 2475, RSP02441 | FT | CIB2; CIB4; CIB5 |
| E-box | 2645, RSP02611 | At3g14205 | bHLH122 |
| E-box | 2647, RSP02613 | ERF6 (At4g17490) | bHLH122 |
| E-box | 2649, RSP02615 | ERF6 (At4g17490) | bHLH122 |
| ANAC089 BS 2 | 2664, RSP02630 | sAPX | ANAC089 |

Gap box 2 (TTTTCAT)—a regulatory element located in the regulatory region of the chloroplast localized glyceraldehyde-3-phosphate dehydrogenase (GapB) gene. Gap box binding factor (GAPF) interacts with this element [26]. A dehydrogenase is an enzyme of the oxidoreductase group that oxidizes a substrate by reducing an electron acceptor. Dehydrogenases are important enzymes in plant primary metabolism [27].

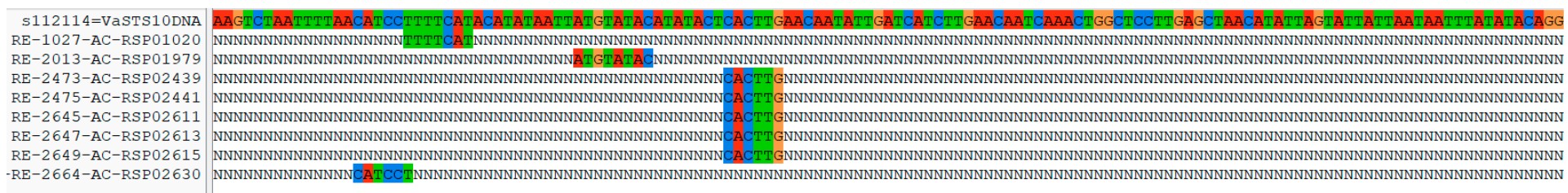

**Figure 4.** Regulatory elements found in the single 135 bp intron of the *VaSTS11* gene using NSITE-PL database [7].

EIN3 BS2 (ATGTATAC)—a regulatory element located in the regulatory region of the ethylene-insensitive 3 (EIN3)-binding F box protein 2 (EBF2) gene of *A. thaliana*, which is a part of the Skp, Cullin, F-box containing complex (SCF complex). SCF is a multi-protein E3 ubiquitin ligase complex that catalyzes the ubiquitination of proteins destined for 26S proteasomal degradation. It is located in the nucleus and is involved in the ethylene-response pathway [28]. EIN3, or ethylene-insensitive 3, is the binding factor for EIN3 BS2, and is a nuclear transcription factor that initiates downstream transcriptional cascades for ethylene responses.

E-box (CACTTG)—a regulatory element located in the promoter region of the flowering factor gene CRY2-interacting bHLH 1, cryptochrome-interacting main helix-loop-helix (CIB1). The CIB1 protein recognizes E-box and activates the transcription of the flowering factor of A. thaliana FT (Flowering locus T) by binding to cryptochrome CRY2 in the presence of blue light [29]. However, representatives of the CIB1-CIB5 proteins do not need the presence of blue light and CRY2 to enhance transcription of the target gene, only the presence of a recognizable regulatory E-box element is sufficient [29].

ANAC089 BS 2 (CATCCT)—a regulatory element located in the regulatory region of the chloroplastic stromal ascorbate peroxidase *sAPX* gene. sAPX scavenges hydrogen peroxide in plant cells [30]. Protein ANAC089 interacts with this regulatory element. NAC—NAM, ATAF, and CUC domain proteins comprise one of the largest plant-specific transcription factor families, represented by ~105 genes in Arabidopsis. ANAC089—one of the proteins of the NAC family transcription factor that negatively regulates floral initiation.

Probably, the presence of described regulatory elements within the *VaSTS11* intron signals to the GAPF, EIN3, CIB, bHLH122, and ANAC089 *A. thaliana* proteins about the need to activate transcription of the *VaSTS11d* transgene. The *VaSTS11c* transgene does not bear such regulatory elements and therefore exhibits a lower expression level in general. This assumption can explain the differences in the level of *VaSTS11* expression, *VaSTS11* cytosine methylation, and stilbene contents in the two analyzed generations of *A. thaliana*.

We also noticed that the flower number was 2–3 times higher in *VaSTS11d*-overexpressing *A. thaliana* than in *VaSTS11c*-overexpressing ones. The analysis revealed that there are regulatory elements both stimulating (E-box) and inhibiting (ANAC089 BS 2) flowering. Interestingly, the number of known E-box-interacting proteins is higher than the number of ANAC089-interacting proteins. Perhaps E-box serves as a signal to accelerate flowering in the intron-containing VaSTS11d-transgenic *A. thaliana*.

GapB—chloroplast localized glyceraldehyde-3-phosphate dehydrogenase; GeneID: 840895 (GAPB), TAIR: AT1G42970. GAPF—Gap box binding factor. EBF2—ethylene-insensitive 3 (EIN3)-binding F box protein 2 of *Arabidopsis thaliana*, part of the Skp, Cullin, F-box-containing complex (SCF complex) is a multi-protein E3 ubiquitin ligase complex that catalyzes the ubiquitination of proteins destined for 26S proteasomal degradation, it is located in the nucleus and is involved in the ethylene-response pathway; GeneID:832606, TAIR:AT5G25340. EIN3—ethylene-insensitive 3, a nuclear transcription factor that initiates downstream transcriptional cascades for ethylene responses; GeneID:821625, TAIR:AT3G20770. FT—flowering locus T, promotes flowering and FT is expressed in leaves and is induced by long-day treatment; GeneID:842859, TAIR:AT1G65480. CIB1—transcription factor, cryptochrome-interacting basic-helix-loop-helix 1. CIB1 interacts with CRY2 (cryptochrome 2) in a blue-light-specific manner in yeast and Arabidopsis cells, and it acts together with additional CIB1-related proteins to promote CRY2-dependent floral initiation. CIB1 positively regulates FT expression; GeneID:829605, TAIR:AT4G34540. CIB2—cryptochrome-interacting basic-helix-loop-helix 2; GeneID:834912, TAIR:AT5G48560. CIB3—cryptochrome-interacting basic-helix-loop-helix 3; GeneID:819922, TAIR:AT3G07340. CIB4—cryptochrome-interacting basic-helix-loop-helix 4; GeneID:837549, TAIR:AT1G10120. CIB5—cryptochrome-interacting basic-helix-loop-helix 5; GeneID:839167, TAIR:AT1G26260. At3g14205—phosphoinositide phosphatase family protein, suppressor of actin 2 (SAC2); GeneID:820638, TAIR:AT3G14205. ERF6 (At4g17490)—ethylene responsive element binding factor 6, encodes a member of the ERF (ethylene response factor) subfamily B-3 of ERF/AP2 transcription factor family (ATERF-6). It is involved in the response to reac-

tive oxygen species and light stress; GeneID:827463, TAIR:AT4G17490. bHLH122—encodes a basic helix-loop-helix-type (bHLH) transcription factor involved in photoperiodism flowering; GeneID:841537, TAIR:AT1G51140. sAPX—a chloroplastic stromal ascorbate peroxidase that scavenges hydrogen peroxide in plant cells; GeneID:826396, TAIR:AT4G08390. ANAC089— NAM (no apical meristem), ATAF (Arabidopsis transcription activation factor), and CUC (cup-shaped cotyledon) or NAC domain containing protein 89, a membrane-tethered transcription factor that negatively regulates floral initiation; GeneID:832289, TAIR:AT5G22290. * http://www.softberry.com/berry.phtml?topic=nsitep&group=programs&subgroup=promoter, accessed on 1 April 2023 [7].

Thus, we analyzed the *VaSTS11* gene, as a representative of the multigenic *STS* family with the shortest intron in *V. amurensis*, introns in other genes are much larger (Figure 5), so they carry even more regulatory elements that may have a different effect on transgene expression.

### 2.5. Stilbene and Biomass Accumulation in the Grapevine VaSTS11-Transgenic Cell Lines

In order to verify the higher effect of plant transformation with the intron-containing *VaSTS11d* on the transgene transcript level and stilbene production in comparison with *VaSTS11c,* we applied the vector constructions with the *VaSTS11d* and *VaSTS11c* to another model system, i.e., agrobacterium-mediated transformation of grapevine cell cultures with the intron-containing *VaSTS11d* and intronless *VaSTS11c* transgenes.

To establish *VaSTS11*-transgenic cell cultures of *V. amurensis*, the V7 suspension culture of *V. amurensis* was incubated with the *A. tumefaciens* strains. After two days in suspension, cefotaxime (250 mg/L) was added to remove agrobacteria, and on the fifth day of cultivation the suspension cells were transferred onto the solid supplemented with the selective antibiotic Km (10–20 mg/L). Then, we selected transgenic callus cell aggregates in the presence of Km for four months and established several Km-resistant independently obtained callus cell lines as described [31]. These calli did not undergo differentiation on the $W_{B/A}$ in the dark. For further analysis, we used the control VC transgenic cell line and six transgenic cell lines independently transformed with the *VaSTS11c* and *VaSTS11d* genes: three *VaSTS11c*-transformed cell lines (11c-1, 11c-2, and 11c-3) and three *VaSTS11d*-transformed cell lines (11d-1, 11d-2, and 11d-3) of *V. amurensis* (Figure 6).

The *VaSTS11c*- and *VaSTS11d*-transgenic cell lines of *V. amurensis* were proved by RT-qPCR for expression of the *VaSTS11* transgenes. All of the *VaSTS11*-transformed cell lines actively expressed the transgenes, except for the 11c-1 line (Figure 6a). The highest expression of the *VaSTS11* transgene was detected in the 11d-3 cell line and the lowest in the 11c-1 line (Figure 6a). In general, the expression of the *VaSTS11* gene was 1.2–1.9 times higher in the 11d lines than in the 11c lines (Figure 6a).

Higher transgene expression in *VaSTS11d*-transgenic lines correlated (r = 0.77) with a higher stilbene content in comparison with the transgene expression and stilbene content in *VaSTS11c* lines (Figure 6a,b). The highest total stilbene content was detected in the 11d-3 and 11d-2 cell lines reaching 7.3 and 8.5 mg/g DW, which were 5.9 and 6.8 times higher than in the control VC cell line (Figure 6b; Table 2). In general, the total stilbene content and production was 1.6–2.6 times higher in *VaSTS11d*-transgenic lines than in *VaSTS11c*-transgenic lines (Figure 6b; Table 2). The enhancement in the total content of stilbenes in transgenic grapevine cell lines was primarily due to an increase in the content of *t*-resveratrol (Table S1).

The callus tissue samples were harvested from the 35-day-old cultures. The data are presented as mean ± standard error (SE) and were evaluated by one-way analysis of variance (ANOVA), followed by the Tukey HSD multiple comparison test performed in Excel using the XLSTAT software, Version 2023, where $p < 0.05$ was considered to be statistically significant.

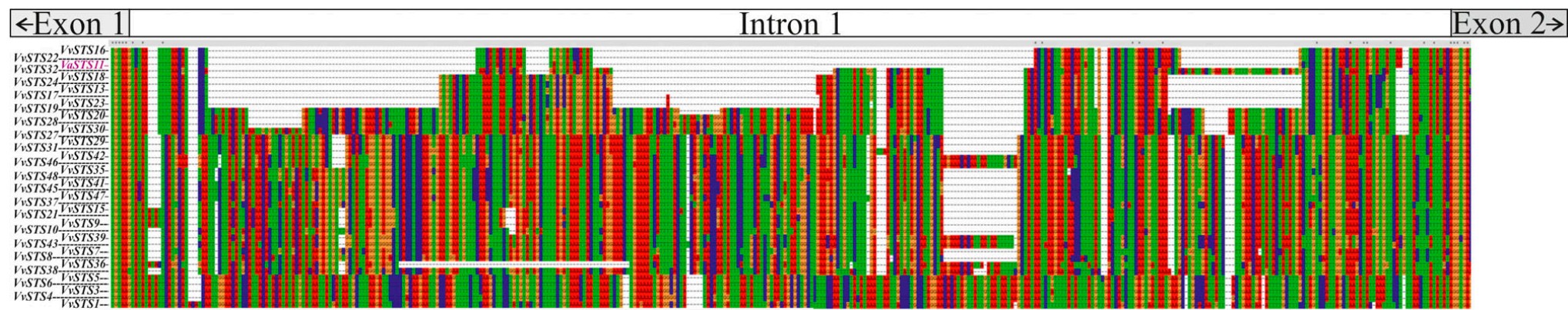

**Figure 5.** Schematic representation of the introns in the 38 functional *Vitis vinifera VvSTSs* genes (except *VvSTS12* and *VvSTS25* due to the lack of intron 1 sequence) and the *Vitis amurensis VaSTS11* gene, a close homolog of the *VvSTS16* and *VvSTS22* genes.

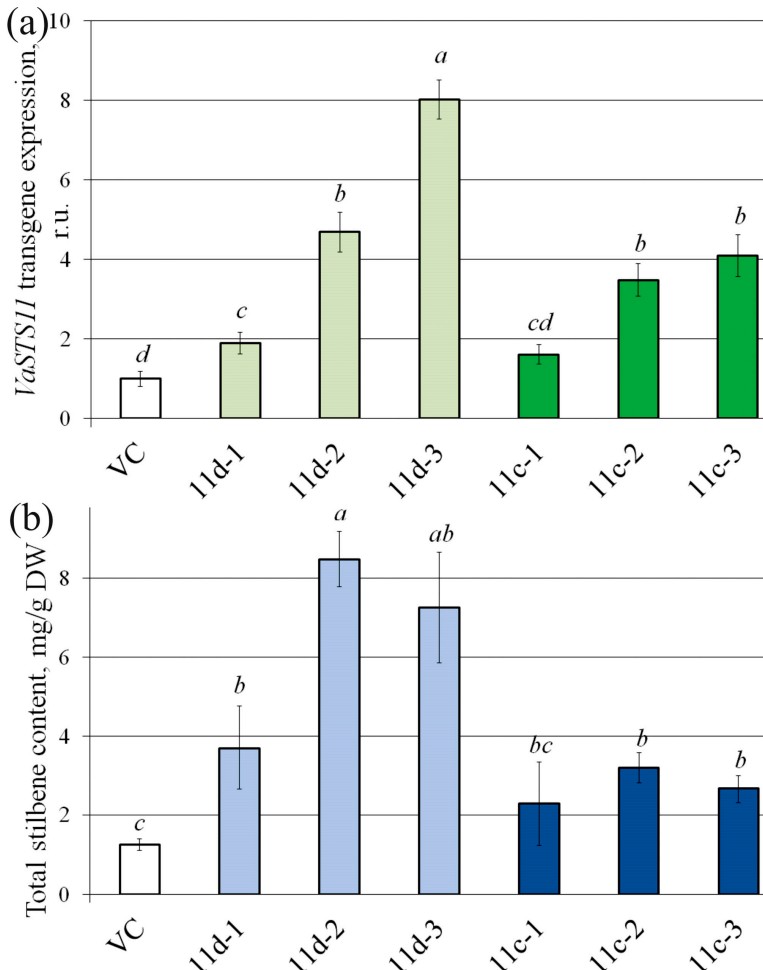

**Figure 6.** Quantification the *VaSTS11* transgene mRNAs performed by quantitative RT-PCR (**a**) and quantification of total stilbene content performed by HPLC (**b**) in the transgenic cell cultures of *Vitis amurensis*. VC—the control *V. amurensis* cell line transformed with the vector harboring only the *npt*II selective marker; 11d-1, 11d-2, and 11d-3—*V. amurensis* cell lines independently transformed with *VaSTS11d*; 11c-1, 11c-2, and 11c-3—*V. amurensis* cell lines independently transformed with *VaSTS11c*; r.u.—related units calculated as described in [32]; fluorescence in VC is indicated as "1". The data are presented as mean ± SE (two independent experiments with eight technical replicates for quantitative RT-PCR and three independent experiments with two technical replicates). Means followed by the same letter were not different using Student's *t* test, where $p < 0.05$ was considered to be statistically significant.

The highest stilbene production was observed in the 11d-3 and 11d-2 *VaSTS11*-transgenic cell lines and reached 77.3 mg/L and 91.8 mg/L, respectively (Table 2). A comparison of the data with the previously published papers revealed that these are the highest values of stilbene levels produced by plant cell cultures overexpressing *STS* genes [19,33–36]. Previously, the highest stilbene production level reached 25.4 mg/L in a cell culture of *V. amurensis* overexpressing the *PjSTS3* gene from spruce *Picea jezoensis* [36], which is 3.6 times less than the content of stilbenes in STS11d-2 and 1.3 times less than the content of stilbenes in the STS11c-2 cell line (Table 2). Thus, grapevine cell cultures also showed better properties of the intron-containing *VaSTS11d* transgene in the activation of stilbene biosynthesis in comparison with *VaSTS11c*.

**Table 2.** Biomass and stilbene accumulation in the cell lines of *Vitis amurensis* overexpressing the *VaSTS11c* or *VaSTS11d* gene transcripts. Means followed by the same letter were not different using Student's *t* test, where $p < 0.05$ was considered to be statistically significant.

| Cell Line | Overexpressed *STS* Gene | Fresh Weight, g/L | Dry Weight, g/L | Total Stilbene Content, mg/g DW | Total Stilbene Production, mg/L |
|---|---|---|---|---|---|
| KA0 | - | 222.1 ± 12.2 [b] | 9.77 ± 0.84 [a] | 1.24 ± 0.17 [c] | 12.1 ± 2.6 [d] |
| 11d-1 | | 260.1 ± 14.7 [a] | 10.98 ± 0.77 [a] | 3.71 ± 1.07 [b] | 40.7 ± 7.5 [b] |
| 11d-2 | *VaSTS11d* | 262.2 ± 15.1 [a] | 10.81 ± 0.92 [a] | 8.49 ± 1.25 [a] | 91.8 ± 8.9 [a] |
| 11d-3 | | 232.4 ± 13.3 [ab] | 10.64 ± 0.65 [a] | 7.26 ± 1.40 [ab] | 77.3 ± 7.7 [a] |
| 11c-1 | | 253.6 ± 15.2 [ab] | 9.81 ± 0.74 [a] | 2.28 ± 1.02 [bc] | 22.4 ± 5.0 [cd] |
| 11c-2 | *VaSTS11c* | 247.4 ± 15.5 [ab] | 9.88 ± 0.98 [a] | 3.21 ± 0.38 [b] | 31.7 ± 5.3 [bc] |
| 11c-3 | | 238.7 ± 16.6 [ab] | 9.84 ± 0.75 [a] | 2.66 ± 0.37 [b] | 26.2 ± 6.1 [bc] |

## 3. Conclusions

Using intron-containing and intronless transgene forms, we found that the intron-containing *VaSTS11d* was better expressed than the intronless transgene sequence (*VaSTS11c*) in $T_4$ and $T_7$ plant generations of transgenic Arabidopsis and in one grapevine cell culture. The data revealed that the function of the plant transgene with an intron was more pronounced since the content and production of stilbenes was higher in the *VaSTS11d*-transgenic Arabidopsis plants and grape cells compared with *VaSTS11c*-transgenic plants. The same results were shown on transgenic Arabidopsis plants, where introducing an intron into a transgene reduced silencing more than four-fold [3].

The active expression of the *VaSTS11d* transgene with the intron was associated with a lower level of transgene cytosine methylation. A reduction in DNA methylation assumed that pre-mRNA splicing involves interactions between the cap-binding complex and components of the spliceosome. These interactions reduce the ability of the spliced transcript to act as a substrate for an RNA-dependent RNA polymerase [3]. Probably, the intron could exert an effect on *STS* expression via an ncRNA which performs gene regulatory functions [6]. Intron-containing genes are suggested to be preferentially channeled to exonucleolytic RNA decay pathways [3,5].

It is possible that the regulatory elements in the intron of *VaSTS11* had a side effect on plant development, since in our case the plants began to flower faster, which can be explained by the presence of the flowering regulatory elements, but this suggestion requires further investigation.

Modern agricultural biotechnology is heavily dependent on using Agrobacterium to create transgenic plants, and it is difficult to think of an area of plant science research that has not benefited from this technology [37]. However, the data of the present study indicate that the transgene lost its properties after the seventh plant generation. Thus, there was no long-term fixation of this gene in the Arabidopsis genome, and this requires separate investigation.

In summary, the data obtained revealed that transgenes with introns have a great potential for use in plant biotechnology, and the results are important for understanding transgene heritage and obtaining vector constructions for effective and stable transgene expression in several plant generations.

## 4. Materials and Methods

### 4.1. Plant Material and Cell Cultures

Plants of *Arabidopsis thaliana* (L.) Heynh. ecotype Columbia-0 (stored by our lab) were grown in pots filled with commercially available rich soil ("Universalniy", Fasko, Moscow, Russia) in an environmental control chamber (Sanyo MLR-352, Panasonic, Tokyo, Japan) kept on a 16/8 h d/night cycle at +22 °C and a light intensity of 120 μmol $m^{-2}$ $s^{-1}$.

For experiments, *A. thaliana* seeds were sterilized for 40–50 min in glass with chlorine vapors, which were released when 3 mL concentrated HCl was added to the 100 mL of bleach (Sayanskhimplast, 7%, Sayansk, Russia). Then, sterile seeds were germinated in Petri dishes in an environmental chamber (+22 °C and 120 µmol m$^{-2}$ s$^{-1}$) on 1/2 Murashige and Skoog medium (MS), pH 5.6 solidified with 0.8% agar. Then, the seedlings grown on the MS medium in Petri dishes for 7–8 days were transferred to commercially available soil.

The V7 callus cultures were established in 2017 from young stems of wild-growing mature *V. amurensis* vines near Vladivostok as described in [38]. The V7 cells were grown for 32–35 days in dark on MS-modified W$_{B/A}$ medium [39] supplemented with 0.5 mg/L BAP, 2 mg/L NAA, and 8 g/L agar in the dark.

### 4.2. Overexpression of VaSTS11c and VaSTS11d in Arabidopsis Plants and Cell Cultures of V. amurensis

The *VaSTS11* transgene was used in two forms, including intronless *VaSTS11c* and intron-containing *VaSTS11d* (Figure 7a). The *VaSTS11d* transgene contained two exons and one intron: exon 1 (E1, 180 bp), exon 2 (E2, 1000 bp), and intron (I, 135 bp). The *VaSTS11c* transgene contained only two exons: E1 and E2 (Figures 1a and S1).

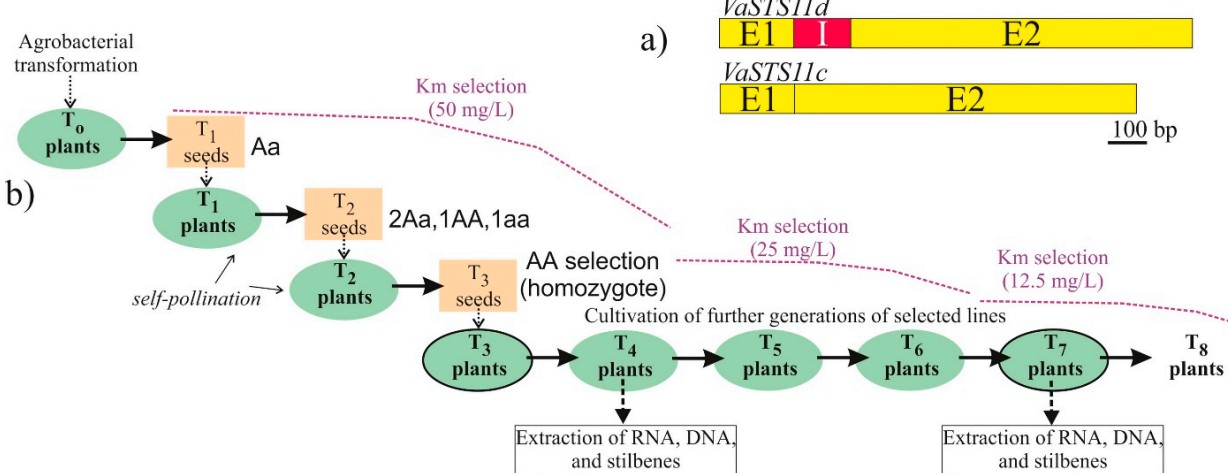

**Figure 7.** (**a**) Schematic representation of the *VaSTS11d* and *VaSTS11c* transgene; E1—first exon; I—intron; E2—second exon. (**b**) Schematic representation of the selection procedure for transgenic *Arabidopsis thaliana*; T$_0$—transformed *A. thaliana* plants; T$_1$–T$_8$—different generations of *A. thaliana* transgenic plants; Km—kanamycin.

To generate the construction for plant cell transformation, the full-length sequences of *VaSTS11d* gene and *VaSTS11c* transcript were amplified by PCR using the primers presented in the Supplementary Figure S1 and Table S2. For amplification of *VaSTS11c* transcript, we used cDNA obtained from a leaf of *V. amurensis*, and for *VaSTS11d* we used DNA from a leaf of *V. amurensis*. We designed primers to the 5′ and 3′ ends of the *VaSTS11d* and *VaSTS11c* cDNA coding sequences based on the known *VaSTS11d* and *VaSTS11c* sequences in *V. amurensis*, respectively (GenBank accession number OQ645979, OQ658380).

The obtained PCR products, *VaSTS11d* and *VaSTS11c*, were subcloned into a pJET1.2 using CloneJET PCR Cloninig Kit (ThermoFisher Scientific, Waltham, MA, USA) and sequenced using an ABI 3130 Genetic Analyzer (Applied Biosystems, Foster City, CA, USA) according to the manufacturer's instructions. Then, we performed PCR with the forward primer containing a *Bgl*II restriction site and the reverse primer containing a *Sal* I restriction site (Supplementary Table S2).

The *VaSTS11d* and *VaSTS11c* were cloned into the pSAT1 vector [40] by the *Bgl*II and *Sal* I sites. Next, the expression cassette from pSAT1 with the *STS* genes was cloned into the pZP-RCS2-*npt*II vector [40,41] using the *Pal*AI (*Asc*I) sites. The pZP-RCS2-*npt*II construction also carried the *npt*II gene. All transgenes in the used vectors were under

the control of the double cauliflower mosaic virus (CaMV 35S) promoter. The overexpression constructs of *VaSTS11* (pZP-RCS2-*VaSTS11d-nptII* or pZP-RCS2-*VaSTS11c-nptII*) or empty vector (pZP-RCS2-*nptII*) were introduced into the of *Agrobacterium tumefaciens* strain (GV3101::pMP90), which was used for the floral dip transformation of *A. thaliana* [18] or for the transformation of the suspension V7 culture of *V. amurensis* [31,38,42].

Two independent fertile $T_3$ homozygous lines of *A. thaliana* transformed with the pZP-RCS2-*VaSTS11d-nptII* (ST11d-1, ST11d-2) or pZP-RCS2-*VaSTS11c-nptII* (ST11c-1, ST11c-2) were chosen for detailed analyses. The transgenic lines used in this study were homozygous plants with a single-copy insertion. We determined the transgene copy number in accordance with the previously published work [43]. We identified the Arabidopsis homozygous lines by germination of all seeds (green seedling) of T3 transgenic Arabidopsis plants on Petri dishes on 1/2 MS (pH 5.6, solidified with 0.8% agar) with the addition of a selective antibiotic Km (50 mg/L).

Additionally, we obtained the control VC transgenic *V. amurensis* cell line and six transgenic cell lines independently transformed with the *VaSTS11c* and *VaSTS11d* genes: three *VaSTS11c*-transformed cell lines (11c-1, 11c-2, and 11c-3) and three *VaSTS11d*-transformed cell lines (11d-1, 11d-2, and 11d-3) of *V. amurensis*.

The $T_1$-$T_4$ generations of *A. thaliana* were selected in the presence of a selective antibiotic kanamycin (Km) at a concentration of 50 mg/L, but then we had to reduce the Km dose to 25 mg/L, since the $T_5$ and $T_6$ generations exhibited slower growth and did not produce seeds at 50 mg/L of Km. Then, the Km concentration was reduced to 12.5 mg/L for the $T_7$ and $T_8$ generations. Notably, the $T_8$ generation grew slowly and did not produce seeds even at this low Km concentration. Km concentration was not further reduced, since non-transgenic plants have been capable of growing and producing seeds at a Km concentration less than 10 mg/L, i.e., transgene selection would not be achieved.

All further experiments were carried out on the $T_4$ and $T_7$ generations of transgenic *A. thaliana* plants (Figure 7b). We used the $T_4$ transgenic plant generation, since it was the first generation with a lot of seeds and plant biomass, which is necessary for efficient nucleic acid isolation and HPLC analysis. The $T_7$ generation was used because this was the last generation of the transgenic *A. thaliana*, which could grow and produce seeds on Km (12.5 mg/L). The $T_8$ generation of the transgenic *A. thaliana* plants presented small plants (rosettes less than 1 cm), which quickly formed stems with flowers, but these flowers did not produce seeds.

To confirm the elimination of *A. tumefaciens*, we used RT-qPCR of the *VirB2* gene using primers presented in Table S2. The transgenic calli were incubated in 100 mL flasks with 50 mL of the solid MS-modified medium [39] supplemented with 0.5 mg/L BAP, 2 mg/L NAA, and 8 g/L agar in the dark. For biomass accumulation and stilbene analysis, the grapevine cell cultures were incubated at 35-day subculture intervals in the dark at 24–25 °C in test tubes (height 150 mm, internal diameter 14 mm) with 7–8 mL of the medium.

### 4.3. HPLC and Mass Spectrometry Stilbene Analysis

Stilbene levels were analyzed by HPLC with diode array detection (HPLC-DAD) as described [39,44]. The extracts were separated on Shim-pack GIST C18 column (150 mm, 2.1 nm i.d., 3 nm part size; Shimadzu, Japan) on the HPLC LC-20AD XR analytical system (Shimadzu, Japan), equipped with an SPD-M20A photodiode array detector. The mobile phase consisted of a gradient elution of 0.1 % aqueous formic acid (A) and acetonitrile (B). An amount of 1 µL of the sample extract was injected with a constant column temperature maintained at 40 °C.

### 4.4. Nucleic Acid Purification and RT-qPCR

The cetyltrimethylammonium bromide (CTAB)-based extraction was used for total DNA isolation as described [45]. The CTAB-based extraction was used for total RNA isolation as described [46]. cDNAs were produced using the MMLV Reverse transcription PCR Kit with oligo(dT)15 (RT-PCR, Evrogen, Moscow, Russia) as described [47].

The mRNA transcript levels of the transgenes were determined by the $2^{-\Delta\Delta CT}$ method [32] with two internal controls, incuding *AtGAPDH* (NM_111283.4) and *AtEF* (XM_002864638) for Arabidopsis, and *VaGAPDH* (XM_002263109) and *VaActin1* (DQ517935) for grape *V. amurensis* as described [48]. The primers designed for RT-qPCRs are shown in Table S2.

RT-qPCR reactions were performed in volumes of 20 μL using the real-time PCR kit (Evrogen) as described [46,47], containing 1 x Taq buffer, 2.5 mM $MgCl_2$, 0.2 mM of each dNTP, 0.2 μM of each oligonucleotide primer, 1x SybrGreen I Real-time PCR dye, 1 μL cDNAs, and 1 unit of Taq DNA polymerase (Evrogen). Analysis was performed in DTprime 4M1 Thermal Cycler (DNA-technology, Moscow, Russia) programmed for an initial denaturation step of 2 min at 95 °C followed by 50 cycles of 10 s at 95 °C and 25 s at 62 °C.

### 4.5. Statistical Analysis

For the analysis of the *VaSTS* transgene expression, we performed two independent experiments with ten technical replicates (five RT-qPCR reactions normalized to one internal control gene and five RT-qPCR reactions normalized to the second internal gene in each independent experiment). Three independent experiments with ten technical replicates in each experiment were performed for callus tissue weight analysis and three independent experiments with two technical replicates in each experiment for the stilbene analysis. The data are shown as mean $\pm$ standard error (SE) and were evaluated by Student's *t* test or by one-way analysis of variance (ANOVA), followed by the Tukey HSD multiple comparison test performed in Excel using the XLSTAT software, Version 2023, where $p < 0.05$ was considered to be statistically significant.

**Supplementary Materials:** https://www.mdpi.com/article/10.3390/horticulturae9040513/s1, Figure S1: Nucleotide sequence of the *VaSTS11d* transgene, 135-nt intron is highlighted with an underscore and a thick font. Table S1: The content of individual stilbenes (mg per g of the dry weight (DW)) in the transgenic cell lines of *Vitis amurensis* transformed with *VaSTS11d* or *VaSTS11c* gene transcripts. Table S2: Primers used for amplification of *Arabidopsis thaliana* and *Vitis amurensis* cDNAs in PCR.

**Author Contributions:** A.S.D. and K.V.K. performed research design, interpretation, and paper preparation. A.A.A. and O.A.A. performed experiments with cell cultures, RNA and DNA isolations, and data analysis. A.R.S. performed HPLC analysis. A.A.A., Z.V.O. and N.N.N. performed RT-qPCRs and participated in data analysis. All authors have read and agreed to the published version of the manuscript.

**Funding:** This work was supported by the grant of the Russian Science Foundation # 22-16-00078, https://rscf.ru/en/project/22-16-00078/ (accessed on 9 April 2023) (Agrobacterial transformation, HPLC analysis, bisulfite sequencing, real-time qPCR). The collection and maintenance of plant material was carried out within the state assignment of Ministry of Science and Higher Education of the Russian Federation (theme No. 121031000144-5).

**Institutional Review Board Statement:** Not applicable.

**Informed Consent Statement:** Not applicable.

**Data Availability Statement:** The data presented in this study are available within the article and Supplementary Materials.

**Conflicts of Interest:** The authors declare no conflict of interest.

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
