# Peer review of "Influence of the 135 bp Intron on Stilbene Synthase VaSTS11 Transgene Expression in Cell Cultures of Grapevine and Different Plant Generations of Arabidopsis thaliana"

_horticulturae, doi:10.3390/horticulturae9040513_

Round 1
Reviewer 1 Report
This MS describes the effect of an intron in the expression of the VaSTS11 transgene in transgenic A. thaliana plants and grapevine callus cell cultures. The authors showed that the presence of the intron resulted in increased expression of the VaSTS11 that was lost in the T5 to T7 progenies of transgenic A. thaliana plants that was possibly due to increased cytosine methylation of the intronless transgene. Finally the showed that the intron possibly contained regulatory sequences that could result to the increased expression of the intron containing transgene.
As already known the floral dip method usually results in the integration of T-DNA in more than one genetic loci. The T-DNA copy number affects the expression of the transgene. Therefore, without a Southern blotting showing that both the intron-containing and intronless transgenics contain the same number of T-DNA insertions, the difference in the expression can be attributed to putative different number of T-DNA insertions. Therefore, I recommend reconsideration of the MS after major revision. Some minor comments/suggestions can be found in the attached file.

Author Response
We are very grateful to the Reviewer for his/her time and the constructive comments on our work. We incorporated all corrections according to the comments in the manuscript (all changes are visible in Ms word tracking change mode).
1) “Reviewer 1,
As already known the floral dip method usually results in the integration of T-DNA in more than one genetic loci. The T-DNA copy number affects the expression of the transgene. Therefore, without a Southern blotting showing that both the intron-containing and intronless transgenics contain the same number of T-DNA insertions, the difference in the expression can be attributed to putative different number of T-DNA insertions. Therefore, I recommend reconsideration of the MS after major revision.”
- Answer: Thanks for this important remark. Indeed, transgene expression may strongly depend on the copy number of T-DNA insertions. The transgenic lines used in this study were homozygous plants with single copy insertion. We determined the transgene copy number in accordance with the previously published work (Gadaleta et al. 2011). In Gadaleta et al. (2011), it has been shown that real-time PCR can be used a fast, sensitive and reliable method for transgene copy number detection. We included this information in the manuscript (Ms). Please, see line 524-525.
- Gadaleta A, Giancaspro A, Cardone MF, Blanco A. Real-time PCR for the detection of precise transgene copy number in durum wheat. Cell Mol Biol Lett. 2011 Dec;16(4):652-68.
2) Some minor comments/suggestions can be found in the attached file.
- a) Line 18: “the canonical transcript” to “the mature transcript”
- Answer: Corrected.
- b) Line 81: “exon 2 (1000 bp), and intron (135 bp)” correct to “exon 2 (E2, 1000 bp), and intron (I, 135 bp)”.
- Answer: Corrected.
- c) Line 82: “The overexpression construct” to “The overexpression constructs”.
- Answer: Corrected.
- d) Line 83: “of VaSTS11d” correct to “of VaSTS11”.
- Answer: Corrected.
- e) Line 85: “which was followed by the floral-dip transformation” correct to “which was used for the floral-dip transformation”.
- Answer: Corrected.
- f) Line 86: There are no data on how the authors have identified the homozygous lines. A paragraph should be added in the Materials and Methods section.
- Answer: We identified the Arabidopsis homozygous lines by germination of all seeds (green seedling) of T3 transgenic Arabidopsis plants on Petry dishes on 1/2 MS (pH 5.6, solidified with 0.8% agar) with the addition of a selective antibiotic Km (50 mg/l). The necessary information is added to the text. Please, see line 525-562.
- g) Line 245: Why this figure is upside-down?
- Answer: It is mistake. We have fixed this error.
- h) Line 247: “element (RE) in” to “element (RE) found in”.
- Answer: Corrected.
- i) Line 300: “confirmed by qRT-PCR for expression” to “confirmed by RT-qPCR for expression”.
- Answer: Corrected.
- j) Line 320: There is no Wt in the graph, “fluorescence in Wt is indicated as “1”” to “fluorescence in VC is indicated as “1””.
- Answer: Corrected.
- k) Line 334: “that this is” to “that these is”.
- Answer: Corrected.
- l) Line 335: “cultures overexpressed STS genes” to “cultures overexpressing STS genes”.
- Answer: Corrected.
- m) Line 377: “environmental chamber” to “environmental controlled chamber”.
- Answer: Corrected.
- n) Line 398: “The obtained PCR products VaSTS11d” to “The obtained PCR products, VaSTS11d”.
- Answer: Corrected.
- o) Line 398-399: we deleted “(DNA extracted from grape V. amurensis leaves)” and “(RNA extracted from grape V. amurensis leaves) transcripts”.
- Answer: Corrected.
- p) Line 402-403:
- Answer: Corrected. We deleted “after cloning and sequencing the VaSTS11d and VaSTS11c transcripts”.
- q) Line 404: “forward primer contained a BglII and the reverse primer contained a Sal I restriction site” to “forward primer containing a BglII and the reverse primer containing a Sal I restriction site”.
- Answer: Corrected.
- r) Line 405:
- Answer: we deleted “which are underlined”.
- s) Line 409-410: After the last sentence of this paragraph a sentence/reference should be added in order to describe the protocol used for the Agrobacterium transformation.
- Answer: Corrected. Please, see line 519-520.
- t) Line 417: “pZP-RCS2-VaMybs-nptII” to “pZP-RCS2-VaSTS11-nptII”.
- Answer: Corrected.
- u) Line 440: “qRT-PCRs” to “RT-PCRs”.
- Answer: Corrected.
- v) Line 441: - The stoichiometry and cycling conditions should be added
- Answer: Corrected. Please, see line 689-695 in new Ms text.
- w) Line 444: “AtEF” to “AtEF” in ithalic.
- Answer: Corrected.
- x) Line 447-454: I cannot find any association with the VaMyb genes.
- Answer: Indeed, this was our mistake. There are no Myb genes in this manuscript. We removed this description form the Ms text.
Reviewer 2 Report
This manuscript by Kiselev et al studied intron's impact by comparing expression of stilbene synthase VaSTS11: intronless vs. with intron.
Overall, the study of intron impact seems interesting. However, I have some questions and gave some examples for improvement:
Why not study the intron-containing VaSTS11d and the intronless transgene sequence (VaSTS11c) in grapevine Vitis amurensis Rupr.?
"2.1. Genetic transformation and selection of the VaSTS11-transgenic Arabidopsis plants"------This part can be described in Materials and Methods.
"However, the data of the present study indicate that the transgene lost its properties after the seventh plant generation. Thus, there was no long-term fixation of this gene in the Arabidopsis genome and this requires a separate investigation."-----Have you compared with other 35S-constructs in arabidopsis?
"Means followed by the same letter were not different using Student’s t test. p < 0.05 was considered to be statistically significant."------This is confusing.
Figure 5 looks weird.
Author Response
We are very grateful to the Reviewer for his/her time and the constructive comments on our work. We incorporated all corrections according to the comments in the manuscript (all changes are visible in Ms word tracking change mode).
3) “Reviewer 2,
Why not study the intron-containing VaSTS11d and the intronless transgene sequence (VaSTS11c) in grapevine Vitis amurensis Rupr.?”
- Answer:
We studied the intron-containing VaSTS11d and the intronless VaSTS11c transgene sequences in cell cultures of grapevine Vitis amurensis (please, see section 2.5. Stilbene content and biomass accumulation in the grapevine VaSTS11-transgenic cell lines). Grape plants were not used in this work, because it takes much longer to establish transgenic grapevine plants, especially up to the 7-8 generation. According to the calculations, this would take at least 30-40 years, since the seeds of the obtained transgenic grapevine plants can be obtained no earlier than 4-5 years after planting in the open ground. For Arabidopsis, we have spent 3 years to establish the 7th and 8th transgenic plant generation (Arabidopsis produces seeds 3-4 months after planting in the ground). Due to the time limitations, these experiments were not carried out on grapevine or other long-growing plants.
4) "2.1. Genetic transformation and selection of the VaSTS11-transgenic Arabidopsis plants"------This part can be described in Materials and Methods.
- Answer: Corrected.
5) "However, the data of the present study indicate that the transgene lost its properties after the seventh plant generation. Thus, there was no long-term fixation of this gene in the Arabidopsis genome and this requires a separate investigation."-----Have you compared with other 35S-constructs in arabidopsis?
- Answer:
When we performed the literature search for relevant scientific papers, we did not find a study where expression or other properties of a plant transgene would be analyzed up to the 7th or 8th transgenic plant generation. Usually, research has been stopped on the 3rd or 4th transgenic plant generation. Therefore, we could not compare our data with other 35S-constructs in Arabidopsis.
6) "Means followed by the same letter were not different using Student’s t test. p < 0.05 was considered to be statistically significant."------This is confusing.
- Answer: These sentences were corrected to: “Means followed by the same letter were not different using Student’s t test, where p < 0.05 was considered to be statistically significant.”. Please, see new Ms text.
7) Figure 5 looks weird.
- Answer: This was a mistake. We have fixed this error in the revised Ms.
Round 2
Reviewer 1 Report
I would like to thank the authors for addressing all my comments. I recommend acceptance of this MS.